# Farming the Edible Aquatic Snail *Pomacea canaliculata* as a Mini-Livestock

Sampat Ghosh [1], Victor Benno Meyer-Rochow [1,2] and Chuleui Jung [1,3,*]

1   Agriculture Science and Technology Research Institute, Andong National University, Andong 36729, Korea; sampatghosh.bee@gmail.com (S.G.); meyrow@gmail.com (V.B.M.-R.)
2   Department of Ecology and Genetics, Oulu University, 90140 Oulu, Finland
3   Department of Plant Medical, Andong National University, Andong 36729, Korea
*   Correspondence: cjung@andong.ac.kr

**Abstract:** In the present paper, we describe the farming system of *Pomacea canaliculata*, an edible freshwater snail, as it is practiced by a farmer as mini-livestock in the vicinity of Andong in Korea. We visited the snail farm several times in the summer and winter of the year and conducted interviews with the farm manager using a semi-structured questionnaire. The farm is housed in polythene tunnels and uses a tank pen of trench type made up of propylene and measuring 1 m × 2 m × 0.5 m (length × width × height) in size. A regulated inflow of fresh water and outflow of used water was installed, with water level not exceeding 5 to 7 cm. As feed of snails, commercial fish feed is generally provided. The life cycle of the *P. canaliculata* might differ in captivity under the controlled environmental conditions than that of an individual in the wild environment. The farming system of snails, particularly *P. canaliculata*, does not involve high labor-intensive, high capital investment and also does not require high through-put cutting edge technology. In addition to providing nutrient-dense snail meat, establishing a snailery could therefore augment the economic condition of farmers in the poorer regions of the world and encourage sustainability and biodiversity conservation.

**Keywords:** biodiversity; food security; mollusca; sustainability

## 1. Introduction

Various land as well as freshwater snails have been accepted as food among different communities around the world. Several scientific reports demonstrate the nutritional potential of snails in terms of high protein and essential amino acids, minerals, and lower fats [1–7]. Snail gathering is often an important source of livelihood for rural dwellers [8,9], but it is unwise harvesting from the wild because it can hurt the biodiversity of the snail community of a region and, moreover, it does not ensure a continuous supply of edible snails throughout the year. It is therefore obvious that such practice is unsustainable. To cite an example, Yildirim et al. [10] noticed a decline in the population of two edible snail species, namely *Helix pomatia* and *Cryptomphalus aspersa*, due to over-harvesting and increasing use of agro-chemicals. In his response to problems of a similar nature, Gheoca [11] suggested heliciculture as a tool to maintain the edible snails' natural population in Romania. However, harvesting of edible snails from the wild environment, mostly during the rainy season, is common in many parts of the world, including several areas of India, where nourishment is poor or scarce and where the establishment of snaileries for farming mini-livestock could be of benefit to the population (Figure 1).

In the realm of globalization and associated dietary transition, traditional food items are often replaced by easily available processed foods [12,13]. Moreover, because of increasing urbanization, the resulting ecological degradation has negatively impacted natural populations of a great number of animal species [14] and related traditional knowledge associated to them, including snails. Thus, in order to ensure that a food resource, such as snails, is sustainable, it is important to consider farming systems. Some developed as

well as developing countries have already started or adopted snail farming as an option to increase the production of protein-rich food. Although the information on environmental sustainability of snail farms is limited, one exhaustive study on the carbon footprint of heliciculture demonstrated that in comparison with conventional livestock, the production of snail meat was accompanied by less greenhouse gas emission [15]. In addition to providing nutritional benefits, snail farms can also improve the economic situation of the farmer and therefore the region of a country.

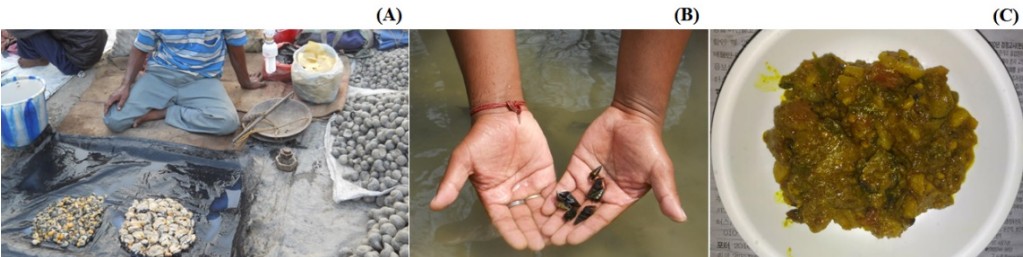

**Figure 1.** (**A**) Selling of edible snails in the northern part of West Bengal, India (Photo credit: A. Sinha Roy); (**B**) collection of edible snails from a pond in West Bengal, India (Photo credit: A. Sinha Roy); (**C**) cooked *Pomacea canaliculata* curry in Korea at S.G.'s house.

In our example, we focus on the farming system of the edible freshwater snail *Pomacea canaliculata*, locally known as "urongi" in Korea. *P. canaliculata* is a snail that belongs to the family Ampullariidae, native to the Neotropical region and as of late has been introduced to Nearctic and Oriental regions as well as a number of oceanic islands. In 1986, *P. canaliculata* appeared in Korea [16]. According to the best of our knowledge and a literature survey, South-East Asia did not have any tradition of snail farming and marketing [16]. As far as Korea is concerned, since the early 1980s, only a few edible snail species have been cultivated, primarily to improve the income of poor farmers [17]. The snails' rich nutrient composition as well as the snail meat's functional properties such as anti-oxidant activities (to cite a few, *Achatina fulica* Bowdich and *Ampullarius insularus* [18]; *Achatina* [17]; *Semisulcospira* [19]; *Pomacea canaliculata* [5]) make snails a potential source of nutrition, be it as an active ingredient to enrich foodstuffs like ready-to-eat 'mandu' with pond snail [20] or a meal by itself as with the European escargot [21].

Therefore, developing a suitable farming system for edible snails could guarantee a regular supply of a highly nutritious food item. Snails produced in a farm would represent a sustainable source and could be a useful tool to mitigate malnutrition, particularly in the underdeveloped or developing world as suggested by Ghosh et al. [22] and recently by Joshi and Pandey [23]. The objective of the present case study is to describe the farming process of *P. canaliculata* comprehensively as practiced successfully by a local farm located in Andong (Republic of Korea) with a hope that the information provided in the article could be helpful for prospective farmers in establishing snaileries especially in the developing and poorer regions of the world.

## 2. Materials and Methods

### 2.1. Farm Site

The snail farm investigated by us is situated close to the city of Andong (36°33′33″ N and 128°43′44″ E) in the Gyeonsangbuk province of the Republic of Korea. Andong, located in the mountainous region of central South Korea, enjoys a temperate climate, with temperatures varying widely in summer and winter between respective extremes of 40 °C and −15 °C. Relative humidity remains within the range of 56 to 80%.

### 2.2. Collection of Information

We visited the snail farm four times in the year (in summer and winter) and conducted interviews with the farm manager and workers using a semi-structured open ended

questionnaire. The information about the *Pomacea canaliculata* farming system as reported in this paper is based on these interviews conducted all four times and our observations during the visits to the snailery. Overall, the semi-structured questionnaire primarily was about the practice of snail farming related to the establishment, life cycle, feed, and nursery management and is presented in Appendix A.

We obtained permission from the farm manager to take photographs of the different components of the farming systems, e.g., house, pen, and tools required for rearing.

### *2.3. Ethics Statements*

The present article describes general practice of snailery and does not involve any experimental procedure on animals or does not carry out any experiment with the animals.

### **3. Results**

### *3.1. Housing*

The snail farm we are describing here is intensive in nature, spread across an area of about 1.65 ha. The farm is housed in polythene tunnels, which serve as green houses with a climate controlling system (Figure 2). The temperature is maintained within the range of 27 to 30 °C. However, in the summer, the farm also uses the open farming system as long as the outside temperature remains favorable (Figure 2).

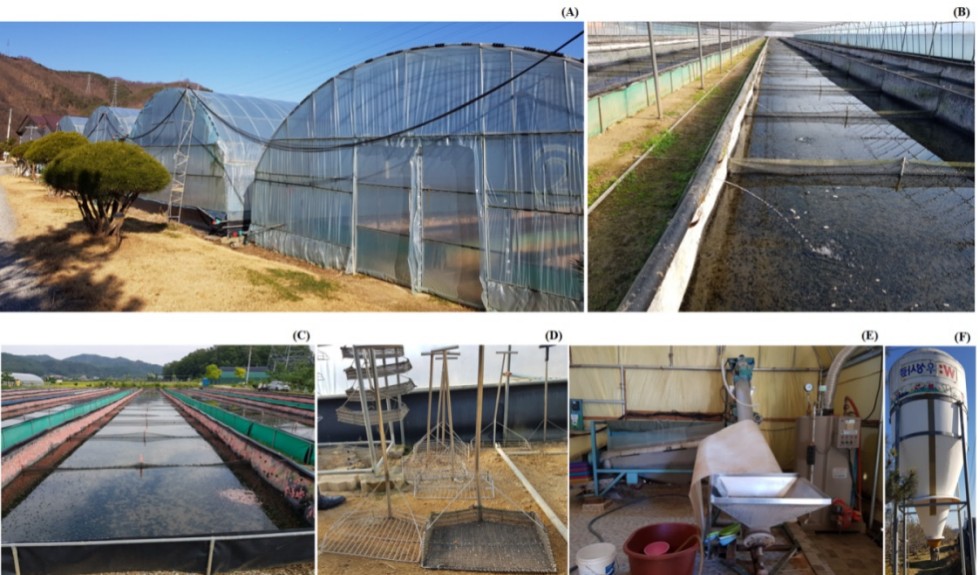

**Figure 2.** Snail farm in Andong (Republic of Korea): (**A**) Tunnel house for *Pomacea canaliculata* snail farming in winter; (**B**) pen for snail rearing; (**C**) open farming system during summer; (**D**) differently-sized steel mesh sheets used to separate snails based on size; (**E,F**) feed manufacturing unit of the farm.

### *3.2. Pen*

The place used for rearing the snails is termed the pen. Here in Korea, the farm we investigated was rearing the freshwater snail *Pomacea canaliculata* using tank pens of the trench type (adjoining snail pens either dug into the ground or raised above the ground) (Figure 2). Each pen was made up of propylene and measured 1 m × 2 m × 0.5 m (length × width × height) in size. A regulated inflow of fresh water and outflow of used water was installed, with water level not exceeding 5 to 7 cm. Mostly surface water from the river and sometimes groundwater has been used on the farm. Water temperature was maintained within the range of 25–29 °C. Water was not allowed to be stagnant to avoid contamination with organic matters like feed, excreta of snails, etc. However, the flow rate of the water supply was very slow, water turnover in the pen took approximately 12 hours to be replaced with new water. There was no significant difference in the nitrogen, carbon,

and phosphorus content of inlet and outlet water, as the farm manager reported. In order to avoid accumulation of snails near the outlet, the pen can be divided by nylon mesh. In this context, it is worth mentioning that care has to be taken not to introduce pests including insects and other small, harmful invertebrates.

### 3.3. Feed

Like all livestock, feed plays a critical role in snail farming. The farm provided commercial fish food known as Punira Aquafeed (containing 23% protein, 3% fat, 10% fiber, 1% calcium, and 1.8% phosphorus) for rearing *P. canaliculata*. However, there was no difference in the feed depending on the life stages of the snail as the farm manager informed us.

### 3.4. Biology and Productivity

Reproductively mature *P. canaliculata* are transferred to the breeding pen. Environmental conditions, especially temperature, play crucial roles in the life cycle of the snail. Growth is highly dependent on ambient temperature, humidity, snail density, and feed. High temperature (more than 30 °C) and low relative humidity (less than 60% RH) slow down the growth of immature snails. *P. canaliculata* lays eggs just 2 months after they hatch, provided the temperature is maintained within the range of 27 to 30 °C. In general, the snails lay eggs once every fifteen days and each clutch or clusters of eggs contains about 100–200 eggs (Figure 3).

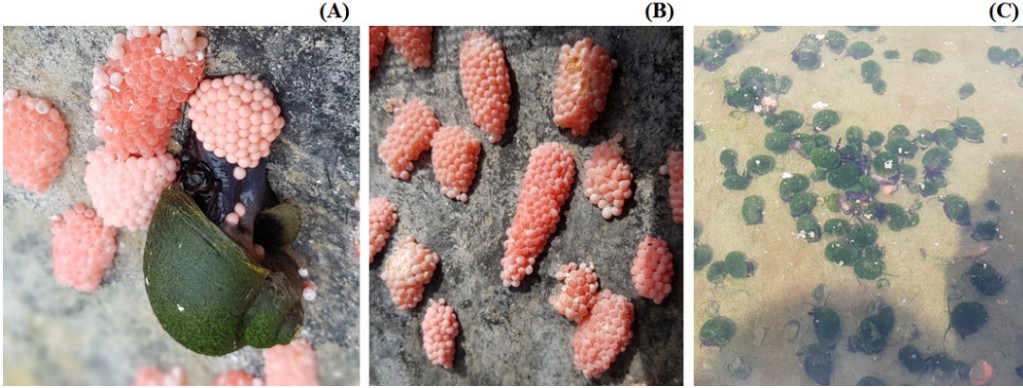

**Figure 3.** (**A**) Snail in the process of laying eggs; (**B**) cluster of eggs attached to the pen wall surface above water level; (**C**) mature snails in the pen.

Normally, the snails lay eggs on the wall of the pen above the water level. In the same environment mentioned before, it takes 7–10 days for the young snails to hatch. In the farm, we found the rate of hatchability to be 80% and the establishment rate (survival after hatching) was 80 to 90%. The hatchlings are collected and kept in different pens. Farmers separate them based on size rather than age using a mesh and keep differently-sized snails in different pens. The density of the snails in a pen is determined by keeping in mind that one snail requires one square centimeter of area. Overcrowding may cause reduced growth, sickness and increased mortality.

Irrespective of the sex, it takes a snail 1.5 to 2 months to become sexually active and reproductively mature. The reproductive period of *P. canaliculata* generally lasts from 2 months to 3 years. The basic life cycle and developmental stages of *P. canaliculata* are shown in Figure 4. Based on their shell height measurement of the different developmental stages rather than specific ages, the snails are categorized and kept in different pens.

The production of the edible snail *P. canaliculata* has been estimated as 6.3 kg (with shell) per square meter area per cycle. As the farm is located in a temperate region, it produces two cycles in a year. As the farm operator further mentioned, in warmer locations, 3 to 4 cycles of production are feasible per year.

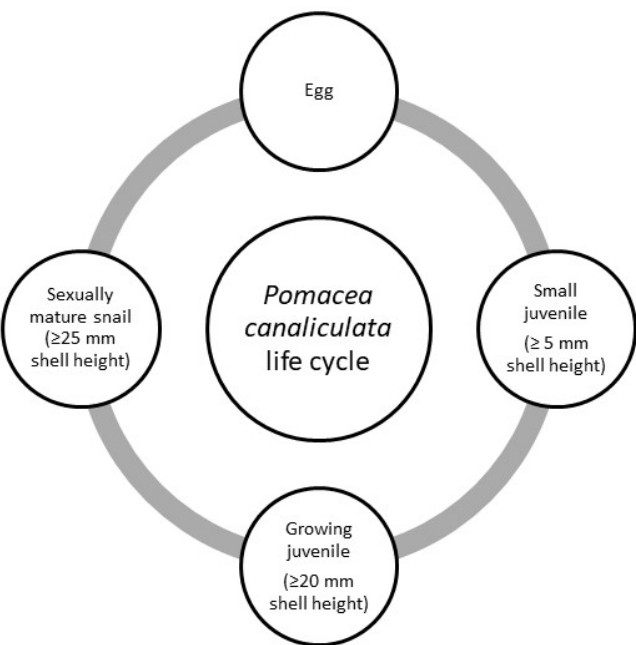

**Figure 4.** Life cycle of *Pomacea canaliculata* based on information obtained from a snail farm in the Andong region of the Republic of Korea.

## 4. Discussion

Site selection is the foremost crucial issue for establishing a successful snail farm. In principle, plain land with humus soil and an adequate amount of shade is recommended as a site to establish a snail farm. In the construction of the snailery, the next important issue is housing, which is largely dependent on the nature of the snail farm, i.e., which species are to be reared. The snail farm can be extensive (which refers to an outdoor free ranging snail pen), semi-intensive (in which egg laying and hatching occur in a controlled environment and the young snails are generally removed to outside pens for further growth or fattening), or intensive (which denotes a closed system throughout the snail's life cycle). However, as many snail species are agricultural pests and may have an extensive negative impact on crops, intensive farming systems could avoid the hazard of escaping snails and spreading in the agriculture fields nearby.

*P. canaliculata* is regarded as one of the worst pests for crops, especially for paddies [16,24]; the intensive farming could prevent harm to the paddy fields located near the farm. Therefore, regardless of the size of the farm, the housing system should meet criteria such as being escape-proof, spacious, and well protected from pests such as insects, predators, etc. In the case of land snail farming in tropical areas, the timber used to build the house and/or pen should be termite-resistant wood. Moreover, in temperate regions an extensive or semi-extensive farming system is not possible throughout the year, due in particular in the winter season.

Depending on the nature of the snail species to be reared, i.e., freshwater (aquatic) or land snail species, pens are of various types such as fenced pens, tank pens, baskets, drums, and trampolines [25]. Although the tank pens of the trench type were used in the farm we had examined and in which *P. canaliculata,* a freshwater species was reared, it is important to discuss the uses of other types of pens. It is worth mentioning that the trench type of pen, generally used in both semi-intensive as well as intensive farming systems, can be used for a variety of purposes and can serve as a hatchery nursery and be used in the fattening process as well.

In the semi-intensive snail farming system, a hutch box is used for breeding purposes. Hutch boxes are rectangular (or square) single or multi-chamber wooden boxes with lids. The boxes, which are kept under regular supervision and protection, are preferably placed into an area that has a climate control system installed [26]. Hutch boxes have small holes

in the bottom and are filled with sieved soil 18 to 25 cm deep. Mature snails from the snailery are transferred to the hutch box and prepared for laying eggs. The breeding snails are kept to their own pens after hatchings start to emerge. The soil of the hutch box should be changed occasionally; otherwise, the increasing amount of droppings can cause disease and unwanted fungal growth.

The third breeding system operates with mini-paddock pens, which are small rectangular (or square) pens usually within a larger fenced area. The scope of using mini-paddock pens exists for the farming of land snails, especially for the purpose of fattening. Mini-paddock pens may not necessarily be small (despite their name), but can be as large as 10 m × 20 m) and may be planted with shrubs or trees to provide shade and shelter from wind, sun and rain. Mini-paddocks as large as these may then be known as free-range pens [26].

Like any other conventional animal husbandry, feed is also an important issue for snail farming. Although the farm used commercial feed, prospective farmers may have a wide option for snails. Snails reared in snaileries are generally herbivores and accept a wide range of food. However, it is of the utmost importance to avoid including plants that produce toxic chemicals in the snails' food. Tender leaves (cassava, pawpaw, eggplant, cabbage, lettuce, etc.), fruits (banana, pawpaw, mango, pear, palm, tomato, cucumber, etc.), tubers (yam, cassava, sweet potato, plantain etc.), or even some household waste (peel of fruits and tubers, leftovers of cooked rice, beans, etc.) are preferred as feed to rear snails with. One can also prepare feed for snails including soybean meal, fish meal, wheat meal, bone meal, premixed vitamins and minerals, etc. Since snails are generally nocturnally active animals, it is preferable to supply them with feed before nightfall and to remove uneaten food in order to maintain sanitary conditions.

Unlike many snail species, *P. canaliculata* is not a hermaphrodite [27]. Controlling the climatic condition is practiced by modern snaileries including the studied farm, but if that is not an option, farming should preferably start at the onset of the wet season. To start the snailery, farmers are required to collect the snails from the wild and fatten them up in captivity. Farmers can, of course, also purchase their breeding stock from suppliers, if the latter are available. However, the most reliable procedure to obtain the breeding or parental stock initially is to obtain them from known breeders or agricultural institutes. This practice continues until the farm becomes self-sustaining.

Once a farm has become established, it can select its breeding stock. In the selection of the most suitable breeding stock, five parameters are to be considered viz., fecundity (number of eggs), hatchability (percentage of eggs likely to hatch), establishment rate (percentage of snails likely to survive after hatching), growth rate (days taken to be sexually mature), and shell strength. As expected, in the farm the hatching rate was found higher than that reported for *P. canaliculata* in its natural habitat (51.6%) and the hatching period was reported to be within the range of 10.3 to 16.5 days [28]. However, scientific studies have revealed that environmental conditions such astemperature seriously influence the life cycle of *P. canaliculata*.

In a study from Argentina, it was reported that *P. canaliculata* survived less than 1 year and showed semelparous behavior (reproducing only for one season) when kept at a constant temperature of 25 °C, but that the species lived for up to 4 years and exhibited iteroparous behaviors when kept at variable temperatures [29,30]. On the other hand, Yoshida et al. [26] reported that despite variable temperature conditions, in Japanese paddy fields, the species appeared to be semelparous. In its natural habitat, *P. canaliculata* (like most other non-marine snails) face two stressful periods, namely cold temperature during the winter and dry conditions in the summer [30]. These stressful periods for the snail can be overcome in intensive production systems by using climate control equipment.

As Andong (Korea) is located in a temperate region and the polythene tunnel houses were built for intensive farming, the problems of insect pests was comparatively small. However, in the tropical region, it could be high. In order to protect the farm from intruding pest insects, the snail farmer usually digs trenches all around the bottom of the fence and

applies used engine oil to them. Moreover, in order to protect the snailery and the human workers looking after the snails from insects such as mosquitos, nylon mesh or mosquito netting may be used to cover the pen. When setting up a snailery in developing countries, discarded and thoroughly cleaned oil drums or water tanks may be used as simple pens. Oil drums or tanks used as pens for rearing land snails should have some small holes in the bottom for drainage, filled with soil 7 to 10 cm high, and covered with nylon mesh for aeration.

Another concern is food safety. Snails often function as intermediate hosts to a variety of parasites harmful to humans and animals [31]. To cite an example, *Angiostrongylus cantonensis* is a parasitic nematode for which *P. canaliculata* could be a suitable host [32]. In addition to maintaining scientific hygiene and sanitation in the farm, the processing of edible snails (including *P. canaliculata*) is important to avoid contamination. Several studies reported that the consumption of raw or inadequately cooked *P. canaliculata* could cause eosinophilic meningitis if the snail is infected by *A. cantonensis* [33,34]. However, proper processing including washing and proper cooking can easily avoid the risk of contamination. Hollyer et al. [35] suggested that thoroughly cooking of snails (potential host of *A. cantonensis*) to an internal temperature of 165 °F (or 74 °C) could prevent the contamination of the nematode.

## 5. Conclusions

Snail farming and the establishment of a snailery as in mini-livestock farming does not require highly specialized equipment, complicated techniques, sophisticated knowledge, or high economic outlays. Establishing snaileries in developing countries and other poorer regions of the world may not only provide a boost to the nutrition of the residents but can also improve the economic situation of the local farming community. However, the environmental footprint of *Pomacea canaliculata* snail farming in the context of sustainability issues and future effects remains to be examined and is a challenge for future research. However, as *P. canaliculata*, like most other snail species, is a pest for various agricultural crops, proper care should be taken so that snails should not be released from the snailery and spread out to agricultural fields nearby.

**Author Contributions:** Conceptualization, S.G. and C.J.; methodology, S.G.; software, S.G.; validation, S.G., C.J. and V.B.M.-R.; formal analysis, S.G.; investigation, S.G.; resources, C.J.; data curation, C.J.; writing—original draft preparation, S.G.; writing—review and editing, V.B.M.-R.; visualization, S.G. and C.J.; supervision, C.J.; project administration, C.J.; funding acquisition, C.J. All authors have read and agreed to the published version of the manuscript.

**Funding:** This research was funded by BSRP through NRF (National Research Foundation of Korea), Ministry of Education, grant number NRF-2018R1A6A1A03024862.

**Acknowledgments:** Authors sincerely acknowledge the snail farm in Andong (Korea) and farm owner Soon-Gab Kwon for information and allowing us to take photographs of his establishment. We also extend our acknowledgement to A. Sinha Roy for providing a few photographs.

**Conflicts of Interest:** The authors declare no conflict of interest.

## Appendix A. Semi-Structured Questionnaire

Structure and establishment of the farm:

1. The area of the snail farm.
2. Housing and environmental conditions, i.e., temperature and humidity.
3. Structure and measurement of pens for snail breeding and rearing.

Life cycle and events of *P. canaliculata*:

1. Reproductive age of snail.
2. Fecundity, i.e., number of eggs laid by a snail at a time.
3. Is there any seasonal effect on egg laying?
4. Time required for hatching from egg.



5.  Hatchability, i.e., % of eggs likely to be hatched out of total number of eggs.
6.  Time taken for hatching.
7.  Establishment rate, i.e., % of hatchlings likely to survive.
8.  Growth rate in domesticated conditions.

Nursery management:

1.  Requirement of feed (if any difference in feed during different stages of life).
2.  Requirement of water.
3.  Management of hatchling. Do they require being kept separately?
4.  Optimum density of snails in the rearing pen.
5.  Productivity of the farm.

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
