# Peer review of "Farming the Edible Aquatic Snail Pomacea canaliculata as a Mini-Livestock"

_fishes, doi:10.3390/fishes7010006_

Round 1

Reviewer 1 Report

the article is well written but is necessary to describe in more detail some following aspects . The technique is not very innovative but can be easily replicated.

1. In the keywords there are “food safety” and “nutrition” that in the article are poorly treated; perhaps appropriate to specify “animal nutrition”
2. The article, which describe only one farm, talks about a questionnaire of which should provide more details both in the material and methods and in the results;
3. Describe the inlet and outlet quality of water and any waste management; section 3.3 should be described in the Materials and Methods.
4. Describe, if possible, the productivity of this farm compared to other similar structures present in the same geographical area.

Author Response

Thank you for the comments. Details are follow.

Reviewer 1:

Open Review

English language and style

(x) Extensive editing of English language and style required
( ) Moderate English changes required
( ) English language and style are fine/minor spell check required
( ) I don't feel qualified to judge about the English language and style

Yes

Can be improved

Must be improved

Not applicable

Does the introduction provide sufficient background and include all relevant references?

(x)

( )

( )

( )

Is the research design appropriate?

( )

(x)

( )

( )

Are the methods adequately described?

( )

(x)

( )

( )

Are the results clearly presented?

(x)

( )

( )

( )

Are the conclusions supported by the results?

(x)

( )

( )

( )

Comments and Suggestions for Authors

the article is well written but is necessary to describe in more detail some following aspects . The technique is not very innovative but can be easily replicated.

  1. In the keywords there are “food safety” and “nutrition” that in the article are poorly treated; perhaps appropriate to specify “animal nutrition”

We have removed ‘nutrition’ from the list of keywords.

  1. The article, which describe only one farm, talks about a questionnaire of which should provide more details both in the material and methods and in the results.

Thank you for the suggestion. We have included the questionnaire in details in the methods.

‘Structure and establishment of the farm:

  1. The area of the snail farm.
  2. Housing and environmental conditions i.e. temperature and humidity.
  3. Structure and measurement of pens for snail breeding and rearing.

Life cycle and events of P. canaliculata:

  1. Reproductive age of snail.
  2. Fecundity i.e. number of eggs laid by a snail at a time.
  3. Is there any seasonal effect on egg laying?
  4. Time required for hatching from egg.
  5. Hatchability i.e. % of eggs likely to be hatched out of total number of eggs.
  6. Time taken for hatching.
  7. Establishment rate i.e. % of hatchlings likely to survive.
  8. Growth rate in domesticated conditions.

Nursery management:

  1. Requirement of feed (if any difference in feed during different stages of life).
  2. Requirement of water.
  3. Management of hatchling. Do they require to keep separately?’

Optimum density of snails in the rearing pen.

  1. Describe the inlet and outlet quality of water and any waste management; section 3.3 should be described in the Materials and Methods.

Thank you for the suggestion. We have now included the water system in the manuscript now.

‘Mostly surface water from river and sometimes groundwater has been used in the farm. Water temperature was maintained within the range of 25-29ᴼC. Water was not allowed to be stagnant to avoid the contamination with organic matters like feed, excreta of snails etc. However, the flow rate of water supply was very slow, approximately water retained in the pen for 12 hours and replaced with new water. There was no significant difference in the nitrogen, carbon, and phosphorus content of inlet and outlet water, as the farm reported.’

  1. Describe, if possible, the productivity of this farm compared to other similar structures present in the same geographical area.

Yes, we understand and now included. Thank you.

‘The production of edible snail P. canaliculata has been estimated as 20 kg (with shell) per 3 square meter area per cycle. As the farm is located in temperate region, it produces 2 cycles in a year. As they further mentioned, in warmer location 3 to 4 cycle of production is feasible per year.’

Reviewer 2 Report

Dear Authors,

I found your manuscript interesting from a contents point of view, rather well written and presented. It is focused on a current topic, giving new useful insights for the researchers of aquaculture field, related on the studied organism. However, there are some minor and major points to address before the submission, that I briefly summarize as follow.

Introduction section needs more references from lines 46-52. 

Between lines 83-84 you reported: "We visited the snail farm during several times (summer and winter) of the year and conducted interviews with the farm manager using semi-structured questionnaire." Generally, in this kind of studies based on interviews, it is good to report in materials and methods section the questionnaire used, to how many people was administered, times, etc.. All this informations should be reported for a better validity of your data and reproducibility of your study.

Results section is lacking from the point of view of practical data related to the breeding of these organisms. Important informations such as commercial size reaching, commercial value of product, estimation of total production per year linked to analyzed methods, should be reported to give a more accurate idea of this animal farming in Korea to the readers.

Please be sure that all the references are correctly referenced, the style appears uneven in the present form.

Best regards 

The Reviewer

Author Response

Thank you for the comments. Details are followed. 

Reviewer 2:

Open Review

English language and style

( ) Extensive editing of English language and style required
(x) Moderate English changes required
( ) English language and style are fine/minor spell check required
( ) I don't feel qualified to judge about the English language and style

Yes

Can be improved

Must be improved

Not applicable

Does the introduction provide sufficient background and include all relevant references?

( )

(x)

( )

( )

Is the research design appropriate?

( )

(x)

( )

( )

Are the methods adequately described?

( )

(x)

( )

( )

Are the results clearly presented?

( )

(x)

( )

( )

Are the conclusions supported by the results?

( )

(x)

( )

( )

Comments and Suggestions for Authors

Dear Authors,

I found your manuscript interesting from a contents point of view, rather well written and presented. It is focused on a current topic, giving new useful insights for the researchers of aquaculture field, related on the studied organism. However, there are some minor and major points to address before the submission, that I briefly summarize as follow.

Introduction section needs more references from lines 46-52. 

Thank you for this valuable suggestion. We have included suitable references such as Ghosh et al., 2018; Cunningham et al., 2021; Elmqvist et al., 2016.

Between lines 83-84 you reported: "We visited the snail farm during several times (summer and winter) of the year and conducted interviews with the farm manager using semi-structured questionnaire." Generally, in this kind of studies based on interviews, it is good to report in materials and methods section the questionnaire used, to how many people was administered, times, etc. All this informations should be reported for a better validity of your data and reproducibility of your study.

Yes, we understand. We have now included the questionnaire in the methods

Results section is lacking from the point of view of practical data related to the breeding of these organisms. Important information such as commercial size reaching, commercial value of product, estimation of total production per year linked to analyzed methods, should be reported to give a more accurate idea of this animal farming in Korea to the readers.

Please be sure that all the references are correctly referenced, the style appears uneven in the present form.

We have checked. Also, we have added few references.

Reviewer 3 Report

Review for the paper "A brief note on farming of edible snail as mini-livestock, Pomacea canaliculata, practiced in Andong, Korea" by Sampat Ghosh, Victor Benno Meyer-Rochow  and Chuleui Jung submitted to "Fishes".

General comment.

The authors conducted an observational study to describe the main features of a snail farm in Korea. They studied and presented interesting experience concerning farming procedures of Pomacea canaliculata, a species with promising potential for cultivation. This paper is well illustrated and covers relevant literature in this field.

This study is of interest to a wide range of potential farmers as well as scientists focused on new reliable farming species.

Specific remarks.

Line 10. Consider deleting  “this”

Line 12. Consider deleting  “during”

Line 12. Consider replacing “summer” with “the summer”

Line 18. Consider replacing “condition” with “conditions”

Line 45. Consider replacing “P. canaliculata” with “Pomacea canaliculata

Line 46. Consider replacing “the traditional” with “traditional”

Line 60. Consider replacing “is native” with “native”

Line 72. Consider replacing “a highly nutritious food item” with “highly nutritious food items”

Line 84. Consider deleting  “during”

Line 92. Consider replacing “carried” with “carry”

Line 99. Consider replacing “nature of snail” with “nature of the snail”

Line 106. Consider replacing “hectare” with “ha”

Line 111. Consider replacing “In case” with “In the case”

Line 112. Consider replacing “timber used to build the house and/or pen with should be termite-resistant timber” with “a timber used to build the house and/or pen should be termite-resistant”

Line 123. Consider replacing “the breeding purpose” with “breeding purposes”

Line 161. Consider replacing “important” with “importance”

Line 172. “P. canaliculata” should be italicized.

Line 187. Consider replacing “the immature” with “immature”

Line 207. Consider replacing “condition” with “conditions”

Line 215. Consider replacing “like most of” with “like most”

Author Response

Thank you for the comments. Details are followed. 

Reviewer 3:

Open Review

English language and style

( ) Extensive editing of English language and style required
( ) Moderate English changes required
(x) English language and style are fine/minor spell check required
( ) I don't feel qualified to judge about the English language and style

Yes

Can be improved

Must be improved

Not applicable

Does the introduction provide sufficient background and include all relevant references?

(x)

( )

( )

( )

Is the research design appropriate?

(x)

( )

( )

( )

Are the methods adequately described?

(x)

( )

( )

( )

Are the results clearly presented?

(x)

( )

( )

( )

Are the conclusions supported by the results?

(x)

( )

( )

( )

Comments and Suggestions for Authors

Review for the paper "A brief note on farming of edible snail as mini-livestock, Pomacea canaliculata, practiced in Andong, Korea" by Sampat Ghosh, Victor Benno Meyer-Rochow  and Chuleui Jung submitted to "Fishes".

General comment.

The authors conducted an observational study to describe the main features of a snail farm in Korea. They studied and presented interesting experience concerning farming procedures of Pomacea canaliculata, a species with promising potential for cultivation. This paper is well illustrated and covers relevant literature in this field.

This study is of interest to a wide range of potential farmers as well as scientists focused on new reliable farming species.

Specific remarks.

 Line 10. Consider deleting  “this”

Done, thank you.

Line 12. Consider deleting  “during”

Done, thank you.

Line 12. Consider replacing “summer” with “the summer”

Done, thank you.

Line 18. Consider replacing “condition” with “conditions”

Done, thank you.

Line 45. Consider replacing “P. canaliculata” with “Pomacea canaliculata

Done, thank you.

Line 46. Consider replacing “the traditional” with “traditional”

Done, thank you.

Line 60. Consider replacing “is native” with “native”

Done, thank you.

Line 72. Consider replacing “a highly nutritious food item” with “highly nutritious food items”

Done, thank you.

Line 84. Consider deleting  “during”

Done, thank you.

Line 92. Consider replacing “carried” with “carry”

Done, thank you.

Line 99. Consider replacing “nature of snail” with “nature of the snail”

Done, thank you.

Line 106. Consider replacing “hectare” with “ha”

Done, thank you.

Line 111. Consider replacing “In case” with “In the case”

Done, thank you.

Line 112. Consider replacing “timber used to build the house and/or pen with should be termite-resistant timber” with “a timber used to build the house and/or pen should be termite-resistant”

Done, thank you.

Line 123. Consider replacing “the breeding purpose” with “breeding purposes”

Done, thank you.

Line 161. Consider replacing “important” with “importance”

Done, thank you.

Line 172. “P. canaliculata” should be italicized.

Done, thank you.

Line 187. Consider replacing “the immature” with “immature”

Done, thank you.

Line 207. Consider replacing “condition” with “conditions”

Done, thank you.

Line 215. Consider replacing “like most of” with “like most”

Done, thank you.

Round 2

Reviewer 2 Report

Dear Authors,

thank you for following my previous suggestion on your manuscript. All the requested revision were done correctly.

Good Luck 

The Reviewer

Author Response

Thank you very much for your valuable advice. We have followed all of your suggestions and revised our manuscript accordingly. Please find the revised manuscript enclosed and our responses below.

The MS was very improved. However, it still needs some revision as follow:

L 99 to 117 Move to apendice or suplementar material

We agree, we have moved the questionnaire to Appendix-A.

L 269 Represents the productivity in kg/m2. Thus it will read 6.3kg/m2.

Yes, thank you for your suggestion, we did it.

L 382 to 384 – Add a discussion on avoiding the risks for consumers caused by eating P. canaliculata (e.g., it should be cooked?)

Yes, we have discussed the processing and food safety issue, mention safe cooking conditions and added two references in this section. Thank you.

Conclusions should address the risks of disseminated P. canaliculata as an agricultural pest or as human diseases vector.

Thank you, we added this point in the conclusion.